# Polyester Polymeric Nanoparticles as Platforms in the Development of Novel Nanomedicines for Cancer Treatment

**DOI:** 10.3390/cancers13143387

**Published:** 2021-07-06

**Authors:** Enrique Niza, Alberto Ocaña, José Antonio Castro-Osma, Iván Bravo, Carlos Alonso-Moreno

**Affiliations:** 1Centro Regional de Investigaciones Biomédicas, Unidad NanoCRIB, 02008 Albacete, Spain; enrique.niza@uclm.es (E.N.); joseantonio.castro@uclm.es (J.A.C.-O.); 2School of Pharmacy, University of Castilla-La Mancha, 02008 Albacete, Spain; 3Experimental Therapeutics Unit, Hospital Clínico San Carlos, IdISSC and CIBERONC, 28040 Madrid, Spain; albertoo@sescam.jccm.es

**Keywords:** nanomedicine, polymeric nanoparticles, drug delivery system, cancer treatment

## Abstract

**Simple Summary:**

Despite the existence of powerful therapeutic agents, cancer is still an incurable disease in many clinical scenarios. In this regard, nanomedicine and particularly polymeric nanoparticles have raised attention as a manner to improved drug delivery. Polymeric nanoparticles can optimize existent compounds or be used to improve the formulation for novel therapeutics. In this article the advantages and disadvantages of polymeric nanoparticles will be discussed, and current nanodevices, raw materials for their formulation, methods of formulation, and polymeric nanoparticles in clinical investigations will be reviewed. Finally, options for improvement and clinical applications will be suggested.

**Abstract:**

Many therapeutic agents have failed in their clinical development, due to the toxic effects associated with non-transformed tissues. In this context, nanotechnology has been exploited to overcome such limitations, and also improve navigation across biological barriers. Amongst the many materials used in nanomedicine, with promising properties as therapeutic carriers, the following one stands out: biodegradable and biocompatible polymers. Polymeric nanoparticles are ideal candidates for drug delivery, given the versatility of raw materials and their feasibility in large-scale production. Furthermore, polymeric nanoparticles show great potential for easy surface modifications to optimize pharmacokinetics, including the half-life in circulation and targeted tissue delivery. Herein, we provide an overview of the current applications of polymeric nanoparticles as platforms in the development of novel nanomedicines for cancer treatment. In particular, we will focus on the raw materials that are widely used for polymeric nanoparticle generation, current methods for formulation, mechanism of action, and clinical investigations.

## 1. Introduction

At the end of the nineties, nanomedicine arose as a panacea for the diagnosis and treatment of diseases. However, today, nanomedicine is still there, waiting for its potential to be fully tapped. 

The history of nanomedicine is linked to the evolution of high-resolution microscopy. The first visualization of <4 nm structures took place in 1902, and was performed by Richard Zsigmondy and Henry Siedentopf. Fifty years later, the disposition of the atoms over surfaces was reported, for the first time, by Erwin Müller in 1951 using ion field microscopy. In this regard, the development of the first atomic force microscope in 1986 allowed us to finally see nanostructures in high resolution [1]. These findings attracted great interest in the field of medicine, and a surge of scientific studies and research allowed us to come up with the term “nanomedicine” [2]. Nevertheless, the first time we saw this concept on paper was in the book “What is Nanomedicine” in 1999, by Robert A. Freiras [3].

Figure 1 illustrates the most important events in the development of nanomedicine as a branch of science. Looking back, we saw the first nanoparticles (NPs) as drug delivery systems (DDS), reported in the late 1960s by Peter Paul Speiser [4]. In 1964, Kulkarni et al. used biodegradable and biocompatible polymers to make NPs for the first time [5]. In the early seventies, Georges Jean Franz Köhler and César Milstein produced monoclonal antibodies (mAbs), which Leserman et al. used to make immunoliposomes years later [6]. This event is considered one of the greatest moments in the history of nanomedicine, because it represented the first “Targeted Nanotherapy”. There were other important findings aimed to implement nanotechnology in the field of medicine, such as the use of dendrimers [7] or chips [8], but, undoubtedly, the use of nanomaterials in tissue engineering paved the way for considering nanomedicine as an area of expertise in science [9]. Aside from this, of particular interest to cancer treatment researchers, is the approval of Doxil^®^ [10] by the Food and Therapeutic Administration (FDA) as the first encapsulated therapeutic, which meant a step forward in the development of new treatments. In fact, it accelerated the registration and marketing approvals of key pharmaceuticals in developed countries [11].

Nanomedicine provides hope to improve current cancer treatment. In this sense, nanoparticles can offer several advantages in comparison to conventional chemotherapeutics based on the enhanced permeability and retention (EPR) effect. The drug can be delivered in high concentrations to the site of interest, reducing the effects to the surrounding tissues (Table 1 collects the advantages of nanomedicines comparted to conventional chemotherapeutics). Apart from all the advantages of the use of DDS for cancer treatment, polymeric DDS are notable for clinical translation, due to the biocompatibility of the raw materials and the easy modulation to improve efficacy.

Nanomedicine can be divided into the following three main areas depending on its application: nanodiagnosis, regenerative medicine, and nanotherapy [12]. The main aim of nanodiagnosis is the early detection of diseases by the use of nanomaterials [13,14,15]. The in vivo diagnosis consists of the administration of different nanodevices for the quantification of several parameters, compounds or metabolites in the organism, while in vitro diagnosis achieves disease detection through samples obtained from patients. One of the principal nanomaterials for nanodiagnosis is the nanobiosensor, which can detect a number of compounds in real time [16].

On the other hand, regenerative medicine consists of the repair or substitution of damaged tissues and organs by nanomaterials [12]. The most commonly used nanomaterials employed for this purpose are based on carbon nanotubes, hydroxyapatite nanodevices, nanocomposites, and biodegradable polymers [17].

Unfortunately, therapeutic agents are not free of side effects and contraindications. In fact, there is a significant proportion of patients who experience adverse effects with the current therapies. Some therapeutics are very quickly metabolized and require high doses to be effective. On the other hand, minor, but frequent, side effects produced mean that many patients report low levels of adherence to treatments [18]. Chemical modifications of approved therapeutics, to improve their pharmacokinetics and safety profile, are costly. In this context, nanotherapy seems to provide solutions by therapeutic encapsulation in controlled-release systems. NPs of 100–400 nm diameter can accumulate within the tumor, through the EPR effect [19]. They can deliver high concentrations of the therapeutic to the target site by convection and diffusion processes, which can also reduce the effects on the surrounding tissues [20].

Figure 2 displays the advantages and disadvantages of the most important DDS (structure, benefits and intended use of the different NPs are described in Table 2). Lipid-based NPs are simply formulated and are able to carry large payloads [21], but they are rapidly retained by the reticuloendothelial system, and modifications to extend their half-time circulation are requested for clinical use. Solid lipid NPs are particularly important in genetic therapy, due to their efficacy for nucleic acid delivery [22]. However, the low therapeutic loading, and accumulation in the liver and spleen limit their options for clinical development [21]. Inorganic NPs possess additional magnetic, electrical, and optical properties for application, such as diagnostics, imaging and photothermal therapies [23]. Nevertheless, their clinical application is limited by their low solubility and toxicity [24]. Amongst the many materials used in nanomedicine, with promising properties as therapeutic carriers, the following one stands out: biodegradable and biocompatible polymers [25,26]. Polymeric NPs are ideal candidates for drug delivery, given the versatility of the raw materials used for their production. These NPs are also stable during storage, and large-scale production is feasible. Furthermore, they show great potential for easy surface modifications to optimize pharmacokinetics, including the half-life in circulation and tissue delivery, and, by modulating the polymer structure, loading and release kinetics can be controlled [27,28,29]. It is worth noting other advantages of polymeric NPs, such as the ease of customized surfaces to be constructed to recognize target proteins and cells [30], along with the generation of stimuli-responsive nanodevices [31,32]. To date, many polymeric NPs are in clinical trials (Table 3) [33].

## 2. From Raw Materials to Polymeric NPs

Polyesters are the most used raw materials for polymeric DDS generation. Ideally, the polymers selected must be biocompatible and biodegradable, and therefore the existence of ester bonds in the macrostructure make these devices easily broken in biological environments. Non-synthetic biodegradable polymers, such as alginate, chitosan, and albumin, have been used to prepare polymeric NPs. Despite their biocompatibility, biodegradability and non-toxic properties, these raw materials present limitations to clinical translation, due to the high variability in batch productions and high immunogenicity of some natural polymers. In addition, their degradation strictly depends on intrinsic properties. On the other hand, synthetic polymers can be designed to modulate delivery parameters such as loading efficiencies, therapeutic release kinetics, surface charge, stability, responsivity, and size and polydispersities of the polymeric NPs. In this sense, polylactide (PLA), polyglycolide (PGA), polycaprolactone (PCL) and polylactide-co-glicolide (PLGA) are commercial FDA-approved polymers for DDS generation [34] (see chemical structures in Figure 3).

### 2.1. PLA

PLA is an FDA-approved polymer, due to its biodegradability, low immunogenicity, low toxicity and high biocompatibility. PLA is degraded to lactic acid, which, in turn, is used in other metabolic routes [35]. Some studies with PLA NPs showed that lactic acid was metabolized fast, to H_2_O and CO_2_, and, therefore, was easily eliminated by the body [36]. PLA is produced by a polycondensation reaction of lactic acid, or by ring-opening polymerization (ROP) of the cyclic ester. Both raw materials are obtained by the anaerobic fermentation of organic substrates such as sugars, beet, or wheat starch [37,38]. PLA has high hydrophobicity, which is an ideal environment for the encapsulation of hydrophobic therapeutics [39]. Representative examples for the development of new cancer treatments using PLA NPs are the work carried out by Coolen et al., where PLA NPs were used for cell transfection [40], or the work reported by Feng et al., to encapsulate fisetin for breast cancer therapy [41].

### 2.2. PGA

PGA is an FDA-approved polymer, obtained by ROP of glycolide. PGA was used for the generation of the first bioresorbable suture in the seventies [42]. PGA is a biodegradable thermoplastic that produces glycolic acid after degradation, and is then excreted in urine. The low solubility in organic solvent, low stability in water, and quick enzymatic degradability limited the use of PGA for NPs formulation. Indeed, the use of PGA is focused on tissue engineering for bone, tendons, cartilage, teeth, and spinal regeneration [43].

### 2.3. PLGA

PLGA is an FDA-approved biodegradable and biocompatible copolymer, obtained by ROP of lactide and glycolide. The degradation products of PLGA are lactic acid and glycolic acid, which are innocuous for humans [44]. The physicochemical properties of the polymer depend on the PLA:PGA ratio. Due to the high degradability of PGA, the degradation of PLGA is marked by the amount of PLA in the copolymer structure [34,44]. PGLA is by far the most used for DDS generation [45]. However, as has happened for all the polymers mentioned, PLGA needs to be pegylated for enhancing in its vivo efficiency [46,47]. The incorporation of polyethylene glycol into the macromolecular structure allows the circulation time of the NPs to increase [48], and the bio-adhesion to different immune cell lines or different plasmatic components to decrease [49,50]. Also, PLGA is used as a raw material for the generation of molecule-targeted therapy. As an original strategy, Pan et al. reported hyaluronic acid-decorated hybrid PLGA nanoparticles as 17-allylaminogeldanamycin delivery carriers for targeted colon cancer therapy. In vivo studies showed much better therapeutic efficiency than the free therapeutic [51].

### 2.4. PCL

PCL is a biodegradable and biocompatible FDA-approved polymer, obtained from fossil resources [52]. This polyester is obtained by ROP of *ε*-caprolactone. It is soluble in a wide range of organic solvents and presents slow degradation rates (2–3 years). Once again, its use is focused on tissue engineering [53]. However, it has also been used as a raw material for DDS generation, in the form of copolymers with other low degradation rate polymers, such as PLGA or PEG [54,55]. There are some successful examples of PCL formulations for the treatment of cancer [56,57,58].

### 2.5. Other Synthetic Polymers

Poly(anhydrides) [59], poly(orthoesters) [60], poly(amides) [61], poly(phosphoesters) [62], and poly(alkylcyanoacrylates) [63] are examples of other polymers used for the generation of DDS. In many cases, successful devices have been formulated, such as the one reported by Fusser et al., using poly(2-ethylbutyl cyanoacry-late) to encapsulate cabazitaxel for breast cancer treatment [64], or the poly(ester amides) NPs reported by Villamagna et al. to encapsulate Celecoxib [65].

## 3. Mechanism of Action of Polymeric NPs

Three mechanisms govern the release of the therapeutic load from polymeric NPs. Burst release, diffusion through the polymer matrix, and erosion are the mechanisms proposed for the release of therapeutics. The extension of each one will depend on the therapeutic structure, raw material employed, NP morphology, and release media [66]. A triphasic profile is expected for the release of the therapeutic from polymeric NPs (see example of profile in Figure 4). Note that mechanisms can act simultaneously.

### 3.1. Stage I

This stage is governed by the burst release mechanism. The therapeutics physically absorbed over the surface of the NPs are released through desorption–diffusion processes. This stage is characterized by very short times.

### 3.2. Stage II

In this case is the diffusion of the therapeutic through the polymer matrix or pores, which explains the release phenomena. This stage is much slower and can coexist with polymer erosion [34,66]. The diffusion relies on the intermolecular therapeutic/polymer forces, as well as the permeability and thickness of the nanodevices [67]. Stage I and II follow the Higuchi pattern, a mathematical model to describe the release of water-soluble and low-soluble drugs incorporated into semisolid and solid matrices. Thus, the simplified Higuchi model describes the release of therapeutics as a square root of a time-dependent process (*t*^1/2^), based on a pseudo-steady state approach that matches the Fick’s 1st law (Fickian release) for an ideal steady-state in stage II [68].

### 3.3. Stage III

The erosion mechanism governs this stage and will depend on the physico-chemical properties of the raw materials [66]. Usually, the release is very slow. Nevertheless, those polymers easily hydrolyzed show this phase in shorter times. In this case, the release kinetic is unpredictable [69]. The Korsmeyer and Peppas model is based on an empirical equation, to include both the Fickian and non-Fickian release of the drug from the polymeric matrix. This describes the release as a time-dependent *t^n^* process, where n is the release exponent, indicative of the mechanism of drug transport through the polymeric matrix, allowing the erosion process to be included in the release mechanism in stage III [68].

Biphasic profiles and constant release profiles have been also reported in a few cases [34].

### 3.4. The Release of the Therapeutic Depends on Many Factors

Cellular and intracellular barriers, crystallinity and degradability of the polymer, the hydrophilic/hydrophobic ratio between the therapeutic and polymer, and the type of surfactants are key variables for the delivery.

#### 3.4.1. Cellular and Intracellular Barriers

Cells and intracellular membranes vary widely, as well as lipid rafts and transmembrane proteins. Therefore, membranes are heterogenous within a patient or across a patient population, which creates diverse barriers to NPs delivery [70]. In addition, polymeric NPs can contact biomolecules such as serum proteins and lipids, to form a corona over their surface. This fact can dictate the uptake of the polymeric NPs, even their stability and cargo [71].

#### 3.4.2. Crystallinity

Polymer crystallinity is defined as the rate of ordered regions inside the polymer matrix in relation with the disordered regions. The orientation of the polymer chains into the NPs can influence the physicochemical properties of the nanodevices. Thus, ordered macrostructures will hamper the water entrance, slowing down the erosion. It is mandatory to calculate the glass transition temperature to know about the crystallinity rate of a polymer (*T*_g_). The *T*_g_ is the temperature in which the polymer changes from a rigid state to a more flexible state, and it is calculated by dynamic scanning calorimetry (DSC) [39]. In general terms, the therapeutic release decreases with the increasing *T*_g_, and the crystallinity is directly related to the molecular weight of the polymer [44]. Polymers with low molecular weights are less crystalline and degrade easily, which causes faster therapeutic release rates [34].

#### 3.4.3. Hydrophilic/Hydrophobic Ratio

This ratio is a limiting factor to achieving high encapsulation efficiencies. Polymers are hydrophobic in nature and, therefore, hydrophobic therapeutics are suited to being entrapped in polymeric NPs. Those polymers with high solubility in water will give rise to DDS with fast therapeutic release profiles [72].

#### 3.4.4. Degradability

Polymer degradation mainly depends on the lability of its chemical bonds, and the crystallinity and hydrophobicity. Polymers with high crystallinity and high hydrophobicity present low degradation rates and low susceptibility to hydrolysis. In the same way, ester and amide bonds facilitate the DDS degradation for enzyme action [66].

#### 3.4.5. Type of Surfactants

Surfactants are humectant, and are capable of reducing the superficial tension of liquids and the interfacial tension between liquids. Each surfactant has a hydrophilic grade, which is a determinant for the generation of NPs [73]. Surfactants can be divided into anionic, cationic and zwitterionic, and non-ionic surfactants. Tweens^®^, spans^®^, pluronic^®^ and poly(vinil alcohol) are the most common surfactants employed in formulation. The surfactants are determined to modulate the size, shape, and surface of polymeric NPs. In general terms, non-ionic surfactants are less toxic. Aggregation phenomena can be observed if the surfactant is not appropriate, impacting the size and polydispersities of the formulation [74]. Also, the release of the therapeutic can be modified [75]. Gagliardi et al. recently discussed the role of surfactants for the construction of polymeric NPs [25].

### 3.5. Systemic Administration of Polymeric NPs Is Common for Cancer Treatment

In terms of size, NPs of less than 10 nm are rapidly eliminated by the kidneys, and those larger than 20 mm stimulate recognition by macrophages or other cells of the immune system [76]. Thus, PEG is usually incorporated into the polymeric NPs to improve their circulation time. Another option is the surface modification by conjugation techniques. Many polymeric NPs have successfully added moieties, such as antibodies, glucose, or proteins, to direct their delivery [77,78].

In terms of surface charge, neutral and slightly negative polymeric NPs have a longer circulation time. Anionic polymeric NPs have to overcome repulsive forces to make contact with the cell surface, and, on the contrary, the attractive forces suffered by cationic polymeric NPs can damage the cells and cause toxicity [79].

The mechanism of action of polymeric NPs is based on the EPR effect. NPs in the range of 100–400 nm have been widely reported to accumulate at the tumor site via the EPR effect [80]. This favors high accumulation of the therapeutic, facilitating its delivery to the site of interest by convection and diffusion processes [81]. However, works suggest potential limitations, because the EPR effect can differ among patients and types of tumors [82,83]. The internalization of the NPs into the cell is produced through endocytosis [82], and results in early endosome formation. The type of endocytosis is determined by cell type and size of the NPs. It should be noted that the endocytosis process can cause changes in the stability of the NPs and their cargo. Figure 5 illustrates the mechanism of action. Endosomes couple to lysosomes that cleave the NPs, which subsequently release the free cytotoxic therapeutics into the cytoplasm, interfering with the cellular mechanisms, and ultimately promoting cell death [84].

## 4. Methods to Formulate Polymeric NPs

There are several methods to formulate polymeric NPs. The methods can be broken down in two main strategies, top-down and bottom-up methodologies (Figure 6).

In top-down methodologies, the NPs are obtained from preformed polymers; meanwhile, in bottom-up methodologies, the polymerization of the monomers is achieved during formulation [85,86]. The nanoprecipitation and displacement solvent method, several techniques of emulsification and evaporation, solvent diffusion, dialysis methods, salting-out, electro-static spraying and micro-fluids are the most important ones in the case of top-down methodologies. Bottom-up strategies have not been widely explored, but, among them, emulsion polymerization, interfacial polymerization, interfacial polycondensation and the coacervation approach are the most used [86]. The following is a more detailed explanation of the most widely used methods for the generation of polymeric NPs (see illustrations in Figure 7).

### 4.1. Simple Nano-Emulsion (Top-Down)

In this approach, the therapeutic and polymer are solubilized in immiscible organic solvents, such as ethyl acetate or dichloromethane, within the aqueous phase containing surfactants [87]. The phases are emulsified with the help of a high-speed homogenizer or sonicator. Once the nano-emulsion is stabilized, the solvent is removed. This methodology is characterized to give rise to large particle sizes [88].

### 4.2. Double or Multiple Nano-Emulsion (Top-Down)

It was designed to encapsulate hydrophilic therapeutics and proteins. This approach consists of the formulation of two nano-emulsions, once a simple nano-emulsion preparation is added to an external aqueous phase, and again emulsified to obtain the double nano-emulsion. NPs are formed when the organic solvent is removed. This approach was designed in order to attain higher encapsulation efficiencies for hydrophilic therapeutics [89].

### 4.3. Salting Out (Top-Down)

This approach is a modified formulation of nano-emulsion in which the mixture to be emulsified contains a polymer, a therapeutic, surfactants, and salting-out agents. The common choices of salting-out agents are magnesium chloride, calcium chloride or sucrose. Fast mechanical stirring is used to emulsify, and the solvent is removed via reduced pressure. The mixture needs ultracentrifugation and repeated washing to eliminate the salting-out agents and surfactants [90]. The main disadvantage of this methodology is that the salting out agents are, in many cases, incompatible with therapeutics [91].

### 4.4. Nanoprecipitation and Displacement Solvent Method (Top-Down)

The polymer and therapeutic are solubilized in miscible organic solvents, and then the mixture is added in a controlled manner over an aqueous solution during continuous stirring [92]. During nanoprecipitation, NPs are formed instantly and the therapeutic is entrapped in the polymer matrix. In this case, the solvent is removed by reduced pressure [28]. The formation of NPs is governed by the Gibbs-Marangoni effect, which describes a mass transfer in an interphase between two fluids, due to a gradient of superficial tension [93]. The nanoprecipitation method prevents the loss of therapeutics during the emulsification process.

### 4.5. Electrosprying (Top-Down)

The basic principles of this approach are based on the application of electrostatic charges to manufacture the NPs. For this approach, a charged solution where the therapeutic and polymer are dissolved is used, and the concentration, caudal, voltage and other parameters are adjusted to generate little drops with different defined shapes and sizes in the matrix solution. This technique achieves very high therapeutic loading efficiency with a low polydispersity index [94].

### 4.6. Microfluids (Top-Down)

The microfluid devices are designed to manipulate fluids in microscale channels. Obtaining NPs in microfluid systems is carried out by microdevices with internal dimensions of less than 1 mm [95,96].

### 4.7. Emulsion Polymerization (Bottom-Up)

Emulsion polymerization is the fastest scale-up method to manufacture polymeric NPs. There are two types [97], emulsion polymerization with a continuous organic phase, which consists of the dispersion of the monomer into an emulsion, and emulsion polymerization, with a continuous aqueous phase in which the monomer is dissolved in an aqueous solution without surfactants. The former is less used because of the use of toxic solvents, surfactants, and initiators, which are difficult to be removed [97].

### 4.8. Interfacial Polymerization (Bottom-Up)

In this case, the mixture of the therapeutic, monomers and initiator are extruded through a needle over an aqueous solution within a surfactant. During the process, NPs are spontaneously formed by monomer polymerization. Later, the solvent is removed, and the NPs are obtained. The advantage of this approach is the high encapsulation efficiency in the one-step formulation. However, the organic solvent is very difficult to remove [97].

## 5. Polymeric NPs in Clinical Investigations

There are more than 15 nanomedicines on the market for cancer treatment [98]. Concerning the polymeric NPs (see Table 3), PICN^®^ is a polymeric formulation of paclitaxel that is approved in India for metastatic breast cancer [99]. The non-targeted PICN^®^ is currently in clinical trials in the USA [98]; Genexol^®^, produced by Samyang Biopharm, is a polymeric micelle formulation of paclitaxel that is clinically approved to treat breast cancer in South Korea [100,101,102]. BIND-014^®^ is composed of a copolymer PLA–PEG and is used for the controlled release of docetaxel against prostate cancer [103,104]. In this particular case, the small-molecule S,S-2-[3-[5-amino-1-carboxypentyl]-ureido]-pentanedioic acid was used to guide the NPs. Preclinical studies showed different pharmacokinetic properties than those reported with sb-docetaxel. Negative phase II clinical trials were reported, due to their little activity [105]. Livatag^®^ is a polymeric NPs formulation of doxorubicin for the treatment of primary liver cancer. Poly (alkyl cyanoacrylate) and cyclodextrin are the raw materials for the generation of this formulation. Initial studies reported higher activities and a very low toxicity in the heart compared to free doxorubicin. However, several adverse pulmonary events led to the termination of the clinical development of Livatag^®^ [106]. CALAA-01^®^ are targeted polymeric NPs generated for siRNA-mediated treatment of solid tumors [107]. They were constructed by the self-assembly of a mixture of cyclodextrin-containing polymer backbone, adamantane-conjugated PEG, and transferrin-conjugated PEG and siRNA. Increased inhibition of tumor growth in mice by CALAA-01^®^ was demonstrated in a mouse model of metastatic Ewing’s sarcoma. It was the first experimental interfering RNA therapeutic agent to be administered in cancer patients [108]. Early results in patients with solid tumors showed dose-dependent intracellular localization in tumor cells. Phase II clinical studies are completed, but no data has been released yet.

The identification of genomic alterations, such as gene amplifications or mutations, in cancers has permitted the design of chemical entities against those alterations. This approach has been shown to be effective, with a wide range of compounds reaching the clinical setting. However, many of those have failed in their clinical development due to an inadequate toxicity profile [109]. The toxic effect of the therapeutic agent against a non-transformed tissue that also expressed the target protein has clearly had a constant limitation: this concept is called on-target off-tumor toxicity [110]. A classic example has been the cardiotoxicity observed with anthracyclines, which is a type of chemotherapy that is widely used for the treatment of many solid tumors. Similar findings can be described for targeted agents, such as the kinase inhibitor against HER2 neratinib that shows an inadequate toxicity profile in relation with diarrhea. In both examples, long-term exposure requires reductions, or even limitations, in their use. Another example is the mucositis and glucose deregulation observed with everolimus, which produces treatment discontinuations [111]. In this context, it is expected that strategies targeting pan-essential genes will be toxic, having an inverse therapeutic index [112,113]. Nano-vectorization is a potential approach to improve their delivery and reduce their toxicity. Another approach will be the identification of smart drug combinations, but, again, both agents could be loaded in the same nanoparticle [114].

In addition, the pharmacokinetic (PK) profile can influence the toxicity and particularly when the toxicity itself is not reversible [109]. To resolve this problem, encapsulation of compounds to improve their PK profile, limiting their exposure to non-transformed tissue, is a main area of research. The encapsulation of PROTACs is an example of success [115], but there is still a long way to go, which requires safety and efficacy experiments in different animal models. Novel methods for the encapsulation of targeted agents, such as small chemical entities, are under evaluation.

## 6. Guided and Smart NPs

Polymeric NPs made of polymers that are naturally pharmaceutically active, or polymers conjugated with therapeutics, or with biomacromolecules such as proteins or antibodies, are being designed to enhance delivery and optimize precision medicine therapies [116]. Therefore, polymeric NPs with targeting moieties, such as antibodies, transferrin or folate, use ligand–receptor, enzyme–substrate or antibody–antigen interactions to improve NP uptake and distribution. Such is the case of the transferrin-targeted polymeric NPs formulated for the codelivery of tarquidar and paclitaxel. These polymeric NPs enhanced cytotoxicity in ovarian cancer cell lines and cancer spheroid cultures [117]. Another example is the conjugation of doxorubicin to PLGA, to form polymeric NPs. The surface of the polymeric NPs was decorated with a hepatocellular carcinoma-specific peptide moiety, to mitigate toxicity in animals and enhance the efficacy of doxorubicin against hepatocellular carcinoma [118]. Another representative example is the nutlin-3-loaded PLGA NPs conjugated with rituximab to successfully target CD20-positive leukemic cancer cells [119], or the paclitaxel-loaded PLGA/montmorillonite NPs conjugated with trastuzumab antibodies, formulated to reduce the side effects of paclitaxel [120]. To exploit this approach, the identification of biomarkers on the surface of tumoral cells is mandatory to avoid on-target non-tumoral toxicity.

On the other hand, smart polymeric NPs can activate depending on the tumor microenvironment or a specific biological stimuli [121]. Some pH-sensitive, visible-light- and temperature-responsive polymeric NPs have been reported to provide successful targeted delivery. A representative example of pH-responsive drug delivery is the ultra-pH-responsive polymers, which are designed for the generation of polymeric NPs for the in vivo delivery of RNAi therapeutics [122]. Interestingly, nanomedicines loaded with paclitaxel and siRNA’s targeting antiapoptotic genes were reported for hepatocellular carcinoma treatment. By the use of low-frequency ultrasound on the tumor site, an enhanced antitumoral activity was observed in vivo [123]. It is worth highlighting that recent technological advances tend to archive multifunctional polymeric NPs that allow the simultaneous accommodation of therapeutic and imaging agents, which, together with the correct choice of the appropriate functionalized smart polymer, facilitate the construction of hybrid multi-modal NPs for theranostic, simultaneous monitoring of drug release and imaging distribution, and to assess therapeutic efficacy in real time [124]. This is the case of the semiconducting polymer nano-PROTACS, which was recently reported for activable photo-immunometabolic cancer therapy [125].

## 7. Conclusions

Over the last decade, polymeric NPs have been designed to overcome the limitations of free therapeutics for the treatment of cancer. Polymeric NPs have shown a more favorable pharmacokinetic profile than the free chemotherapeutics, but optimization of the formulation, in terms of the polydispersity and size of the NPs, is still needed to improve efficacy. In the same way, drug release from polymeric NPs can be more precisely controlled, with a range of polymers designed specifically for that purpose. In this sense, therapeutic polymers emerge as polymers with pharmaceutical and biomedical applications, and with a promising future in cancer research. On the other hand, selective-targeting tumoral cells can augment the permeability and penetration of the polymeric NPs within the tumor, and, therefore, diminish toxicity and avoid the adverse effects of prolonged treatment. For this purpose, guided polymeric NPs may provide benefit over NPs when targeting receptors that also have a biological effect.

In conclusion, although great advances have been performed, there is still a long way to go, and combined efforts from the scientific and biotechnology community are needed in order to speed up the development of this technology.

## Figures and Tables

**Figure 1 cancers-13-03387-f001:**
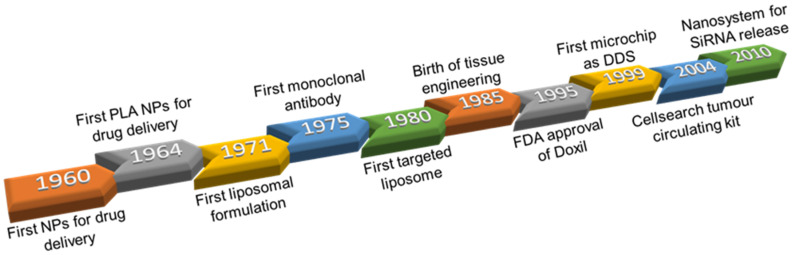
Major discoveries made in the field of nanomedicine, which are linked to the evolution of high-resolution microscopy.

**Figure 2 cancers-13-03387-f002:**
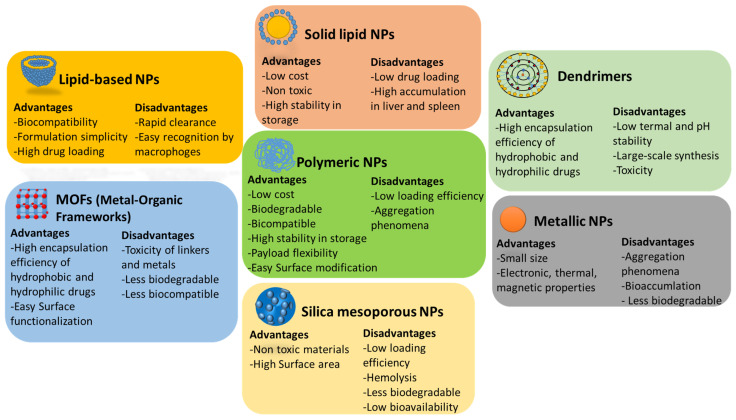
Advantages and disadvantages of different DDS for cancer treatment. Lipid-based NPs reach high drug loading, but their half-life in blood limits their clinical translation; solid lipid NPs fails due to low drug loading and high accumulation; dendrimers are mainly toxic and difficult to scale their synthesis; metallic NPs are easily accumulated in the body and cause aggregation phenomena; MOFs and silica mesoporous NPs are not biodegradable and toxic; polymeric NPs are the most potential candidates for clinical translation due to their high biocompatibility and payload and surface modification flexibility.

**Figure 3 cancers-13-03387-f003:**
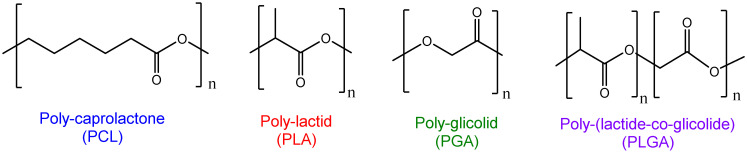
Chemical structures of the most widely used polymers for DDS generation. PCL is used for DDS generation in the form of copolymers with other low degradation rate polymers; PLA is an ideal environment for the generation of polymeric NPs capable of encapsulating hydrophobic therapeutics; PGA is mainly used in tissue engineering; PGLA is by far the most used for DDS generation due to the easy hydrophobic/hydrophilic ratio modulation.

**Figure 4 cancers-13-03387-f004:**
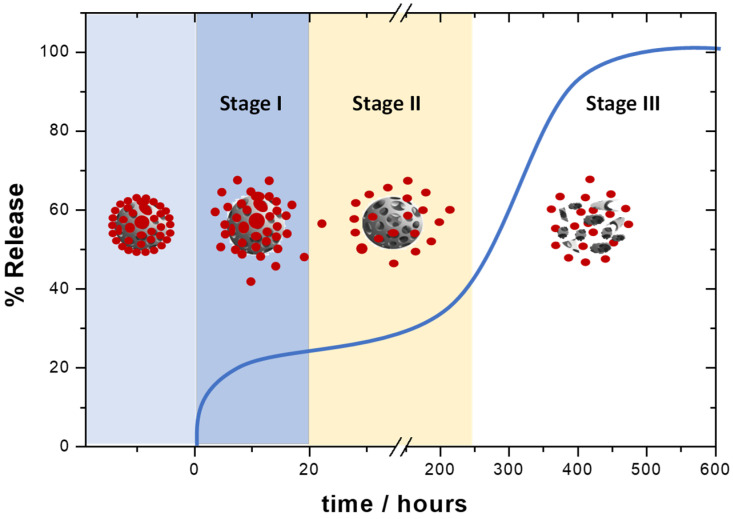
Representative release profile for polymeric NPs. In stage I the drug is quickly released through desorption–diffusion processes. The release of the drug in stage II is governed by diffusion through the polymer matrix or pores. Stage III is governed by the erosion of the polymeric matrix.

**Figure 5 cancers-13-03387-f005:**
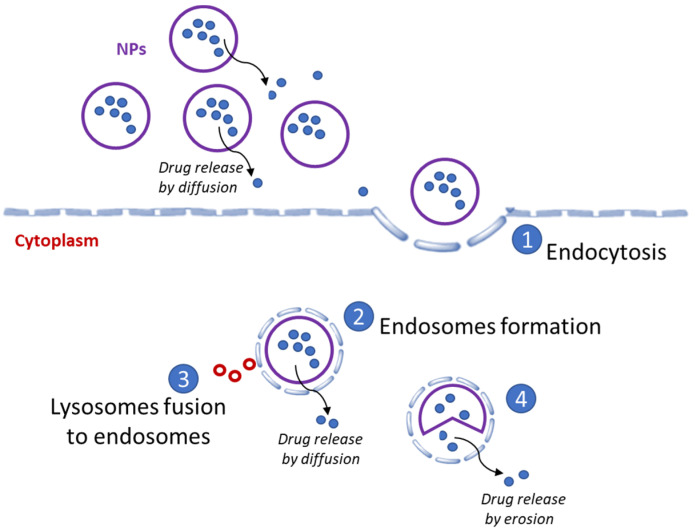
The mechanism of action of polymeric NPs consists of the internalization of the NPs into the cell that is produced through endocytosis, producing changes in the stability and cargo of the NPs. The following step is the cleavage of the NPs by endosome formation and lysosome fusion to release the drug by erosion of the NPs.

**Figure 6 cancers-13-03387-f006:**
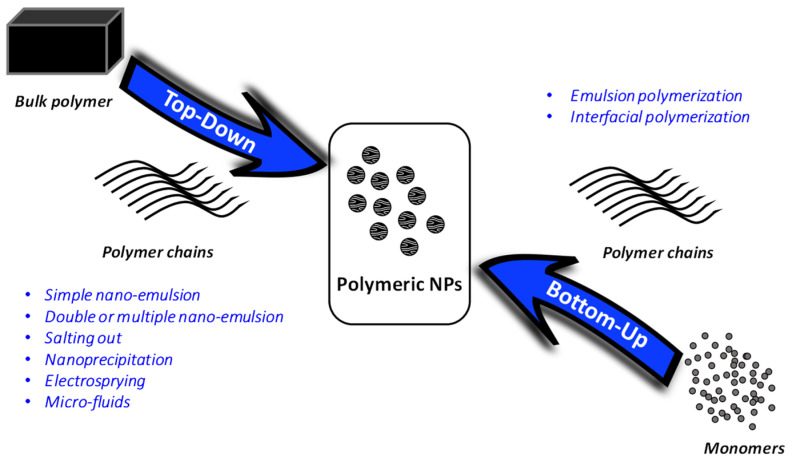
The methods to formulate polymeric NPs are divided into top-down and bottom-up methodologies. Top-down methodologies generate the NPs from preformed polymers, whereas in bottom-up methodologies the polymerization of the monomers is achieved during formulation.

**Figure 7 cancers-13-03387-f007:**
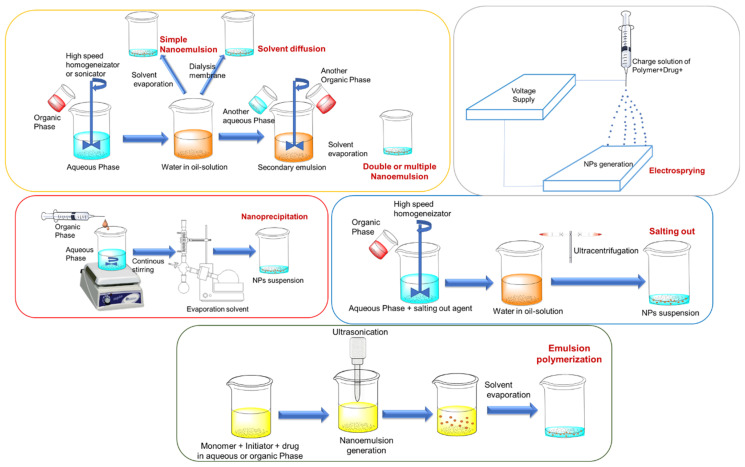
Illustration of the formulation of polymeric NPs by the most significative methods. Emulsion polymerization is a bottom-up methodology, whereas nanoprecipitation, simple nano-emulsion, solvent diffusion, double or multiple nano-emulsion, and salting out are top-down methodologies.

**Table 1 cancers-13-03387-t001:** Advantages of nanomedicine compared to conventional chemotherapeutics.

Advantages of Nanomedicine
Improved bioavailability
Greater dose response
Enhanced solubility
Scope for improvement efficacy
Assess therapeutic efficacy in real time: simultaneous monitoring of drug release and imaging distribution
Reduced toxicity
Allow to design specifically targeted therapies
Possibility of designing smart nanosystems capable of responding to external stimuli

**Table 2 cancers-13-03387-t002:** Structure, benefits and intended use of the most significant DDS in medicine.

Type of DDS	Structure	Benefits	Indented Use
**Lipid-based NPs**	Spherical vesicle having a lipid bilayer	Formulation simplicity and high drug loading	First vehicle for administration of pharmaceutical drugs
**Solid lipid NPs**	Spherical solid lipid core stabilized by surfactants	Temporal and in vivo stability	Delivery vehicle for nucleic acids
**Dendrimers**	Branched polymeric molecules	Encapsulation of hydrophobic drugs	Easy chemical modification to increase in vivo suitability
**MOFs**	Clusters of metal ions coordinated to organic ligands	High encapsulation and loading efficiency	pH-, magnetic-, ion-, temperature- and pressure-response carriers
**Metallic NPs**	Mainly iron oxide, gold, or silver core	Small size and easy surface functionalization	Development of diagnostic and therapeutic agents
**Silica mesoporous NPs**	Mesoporous silica core	Extensive multi-functionality based on its high specific surface	Stimuli-reactive guided liberation of drugs through chemical coatings
**Polymeric NPs**	Biodegradable and biocompatible polymers	Low cost, large-scale synthesis, payload flexibility, biocompatibility and easy surface modification	Clinical translation of nanomedicines

**Table 3 cancers-13-03387-t003:** Polymeric NPs in clinical trials.

Nanomedicine	Drug	Polymer	Conditions	Reference Clinical Trials
**Genexol-PM^®^**	Paclitaxel	polymeric micelle formulation	Metastatic adenocarcinoma of the pancreas	NCT02739633
			Hepatocellular carcinoma after failure of sorafenib	NCT03008512
			Advanced urothelial cancer	NCT01426126
			Advanced non-small-cell lung cancer	NCT01770795
			Advanced, metastatic and recurrent breast cancer	NCT01784120 NCT00876486 NCT01169870 NCT00912639 NCT02263495 NCT02064829
			Gynecologic cancer	NCT02739529
			Advanced ovarian cancer	NCT00877253 NCT01276548 NCT00886717
			Advanced and metastatic pancreatic cancer	NCT00882973 NCT00111904
			Advanced non-small-cell lung cancer	NCT01023347
			Advanced head and neck cancer	NCT01689194
			Advanced esophageal squamous cell carcinoma	NCT01474642 NCT00816634
**PICN^®^**	Paclitaxel	copolymer polylactide-polyehtylene glycol	Metastatic breast cancer	CTRI/2010/091/001116
**BIND-014^®^**	Docetaxel	copolymer polylactide-polyethylene glycol	Metastatic castration-resistant prostate cancer	NCT01812746
			Non-small-cell lung cancer	NCT01792479
			Advanced or metastatic cancer	NCT01300533
			KRAS positive or squamous cell non-small-cell lung cancer	NCT02283320
			Urothelial carcinoma, cholangiocarcinoma, cervical cancer and squamous cell carcinoma of the head and neck	NCT02479178
**Livatag^®^**	Doxorrubicin	Polyalkylcyanoacrylate	Advanced hepatocarcinoma	EudraCT-2006-004088-77
**CALAA-01^®^**	siRNA	Adamantane polyethylene glycol containing cyclodextrin	Solid tumor cancers	NCT00689065

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
