# Peer review of "Polyester Polymeric Nanoparticles as Platforms in the Development of Novel Nanomedicines for Cancer Treatment"

_cancers, 2021, doi:10.3390/cancers13143387_

Round 1

Reviewer 1 Report

The manuscript by Niza, Ocana, Castro-Osma, Bravo and Alonso-Moreno is a review on polyester based nanoparticles as therapeutic delivery system. An overview of fields of interest of the main polyesters used for the generation of DDS, mechanism of action of these polymeric nanoparticles, the main formulation methods and clinical investigation are described. The manuscript is quite interesting, understable and fluent for the reader. Thus, the reviewer recommends publication of this paper after some minor revisions.

1)     The title mentions polymeric nanoparticles but the manuscript focuses only on polyester based nanoparticles. It might be good to modify the title.

2)     L 223 “ff” -> if

3)     L 341 “deregulaion” -> “deregulation”

Author Response

The manuscript by Niza, Ocana, Castro-Osma, Bravo and Alonso-Moreno is a review on polyester based nanoparticles as therapeutic delivery system. An overview of fields of interest of the main polyesters used for the generation of DDS, mechanism of action of these polymeric nanoparticles, the main formulation methods and clinical investigation are described. The manuscript is quite interesting, understable and fluent for the reader. Thus, the reviewer recommends publication of this paper after some minor revisions.

Response: We would like to thank the reviewer for his/her comments and revision of the manuscript.

  • The title mentions polymeric nanoparticles but the manuscript focuses only on polyester based nanoparticles. It might be good to modify the title.

Response: The title has been modified according to the suggestion.

  • L 223 “ff” -> if

Response: The typo has been corrected.

  • L 341 “deregulaion” -> “deregulation

Response: The typo has been corrected.

Reviewer 2 Report

A review by Niza et al focuses on the use of polymer nanoparticles as a platform for the development of new nanotherapeutics. Although the authors tried to present the potential of these nanomaterials as broadly as possible, there is no clinical background here. I describe my comments in more detail below:

1. The main problem, which in my opinion significantly reduces the quality of work, is too much focus on the technical issues of production and use of polymer nanoparticles, and too little background regarding their application in medicine. I believe that the paper would greatly benefit from adding tables describing the use of these nanomaterials in nanomedicine. A short summary describing the type of nanoparticles, their structure, benefits, and intended use would suffice.

2. Simple Summary should be rewritten. I am aware that this is a summary intended for a reader unfamiliar with the subject, however, many terms are too colloquial ("waste of time", "pros and cons") and should not be found in a reputable scientific journal. The situation is similar throughout the article ("pros and cons" - line 88). Please keep the scientific nature of the work by limiting colloquial terms.

3. Line 104 - "To date, many polymeric NPs are in clinical trials" - I think it would be useful here to at least briefly summarize (in the form of a table?) what polymer nanoparticles are in clinical trials and for what purposes they are tested. In the further part of the manuscript, reference is made mainly to those that have either already been approved or tested negative and are not further tested.

4. The descriptions of the figures could be more accurate. Where possible, it would be advisable to add descriptions so that the reader does not have to return to the text (unless he is interested in more precise data) - especially Figures 4-6

Author Response

A review by Niza et al focuses on the use of polymer nanoparticles as a platform for the development of new nanotherapeutics. Although the authors tried to present the potential of these nanomaterials as broadly as possible, there is no clinical background here. I describe my comments in more detail below:

Response: We would like to thank the reviewer for his/her comments and revision of the manuscript.

  1. The main problem, which in my opinion significantly reduces the quality of work, is too much focus on the technical issues of production and use of polymer nanoparticles, and too little background regarding their application in medicine. I believe that the paper would greatly benefit from adding tables describing the use of these nanomaterials in nanomedicine. A short summary describing the type of nanoparticles, their structure, benefits, and intended use would suffice.

Response: Following the suggestions to improve the quality of the manuscript, we have added a table describing the most significant nanodevices designed for cancer treatments. In that table (table 2 in the revised manuscript), we have highlighted the structure of the nanodevice, their benefits and their indented use.

  1. Simple Summary should be rewritten. I am aware that this is a summary intended for a reader unfamiliar with the subject, however, many terms are too colloquial ("waste of time", "pros and cons") and should not be found in a reputable scientific journal. The situation is similar throughout the article ("pros and cons" - line 88). Please keep the scientific nature of the work by limiting colloquial terms.

Response. We agree with this reviewer about the style of the simple summary so far, and, therefore, it has been rewritten. We have avoided the use of words that are too colloquial.

  1. Line 104 - "To date, many polymeric NPs are in clinical trials" - I think it would be useful here to at least briefly summarize (in the form of a table?) what polymer nanoparticles are in clinical trials and for what purposes they are tested. In the further part of the manuscript, reference is made mainly to those that have either already been approved or tested negative and are not further tested.

Response: we have created a table showing the polymeric nanoparticles that are in clinical trials. We have divided the table in different parts such as: the encapsulated drug, the formulation of the raw material , conditions and the clinical trial reference. See Table 3 in the revised manuscript.

  1. The descriptions of the figures could be more accurate. Where possible, it would be advisable to add descriptions so that the reader does not have to return to the text (unless he is interested in more precise data) - especially Figures 4-6

Response: We have added a more detailed description of each figure in the manuscript.

Reviewer 3 Report

This review paper describes the development of nanomedicien and characteristics of polymeric nanoparticles among them. The types and characteristics of representative polyemrs constituting polymeric NPs, the NPs drug release mechanism composed of them, and the characteristics of NPs that affect to release mechanism them, and their fabricating methods was explained in the manuscript.

Comments 1

At the beginning of the manuscript, the development process and explanation of nanomedicine is presented, which is not limited to polymeric nanoparticles. Although this introductory part provides the reader with good information and background, it is recommended to explain more about the advantages of polymeric nanoparticles over other NPs

Comments 2

In the conclusion section, it would be good to briefly summarize the problems that polymeric NPs need to overcome in the future.

Comments 3

I think the “pro-duction” in line 98 should be changed to “production”.

Author Response

This review paper describes the development of nanomedicien and characteristics of polymeric nanoparticles among them. The types and characteristics of representative polyemrs constituting polymeric NPs, the NPs drug release mechanism composed of them, and the characteristics of NPs that affect to release mechanism them, and their fabricating methods was explained in the manuscript.

Response: We would like to thank the reviewer for his/her comments and revision of the manuscript.

Comments 1

At the beginning of the manuscript, the development process and explanation of nanomedicine is presented, which is not limited to polymeric nanoparticles. Although this introductory part provides the reader with good information and background, it is recommended to explain more about the advantages of polymeric nanoparticles over other NPs

Response: In the revised manuscript we have added a short paragraph, just before the section used to classified nanomedice, to explain the benefits of nanoparticles and polymeric nanoparticles in comparison to classical chemotherapeutics and the most significant nanodevices. We have also incorporated a table describing the advantages of the nanoparticles . Furthermore, an additional table (now table 2) has been incorporated to complement the information. In this case the reader can  understand the differences between nanodevices including structure, benefits and the intended use).

Comments 2

In the conclusion section, it would be good to briefly summarize the problems that polymeric NPs need to overcome in the future.

Response: following this suggestion, a critical comment about manners to improve polymeric NPs and their limitations for clinical translation has been added at the end of the conclusion section.

Comments 3

I think the “pro-duction” in line 98 should be changed to “production”.

Response: the typo has been corrected in the revised manuscript.

Reviewer 4 Report

Niza et al., submitted the review entitled “Polymeric nanoparticles as platforms in the development of novel nanomedicines for cancer treatment”.  The authors have summarized the different aspects of polymers NPs, such as type of NPs, design, mechanism of action, methods of preparation. Further, they have also provided brief information about the clinically available polymeric NPs. Figure 1 is an excellent way of representation and timely mannered.

Although the authors have shed a light on different aspects of polymeric NPs, still there a scope to increase the elegance of the manuscript.

Here are some of the suggestions.

I am wondering why the authors have used this abbreviation “therapeutic delivery systems (DDS)” instead of “drug delivery system (DDS)”. Either the authors should use TDS or DDS with respective terminology.

Line 58-60: Not in a flow; please rephrase it.

Line 68: These two references would fit here;

https://doi.org/10.1080/09205063.2021.1909412

https://doi.org/10.1016/j.nantod.2020.101051

Line 73: metabolites (not compounds)

Line 98: production (not pro-duction)

Figure 2:

In MOFs section- Expand “MOFs” in the legend; functionalization (not funcionalization)

Deleted excess “of” in MOFs and denrimers section

Replace “no” with “less” (eg., less biodegradable, less biocompatible, etc)

In metalic NPs section –  “thermal” (not termal)

Section mechanism of action of polymeric NPs - Please change word Phase with Stage. As in the next section the authors discussed the clinical trails readers may confuse with this terminology.

Define the Fick’s law, Korsmeyer and peppas model, and Higuchi equation.

Line 223: if (not ff)

Methods section -  Although they are fundamental methods, they are the real working methods on the bench. Therefore, it would be interesting to illustrate each technique in a pictorial representation either from copyrighted or own designs. This would attract many budding researchers.

Section “polymeric NPs in clinical investigations” – Irrespective of the text, a table with the enlist of drug and polymer information would be attractive and more informative.

Author Response

Niza et al., submitted the review entitled “Polymeric nanoparticles as platforms in the development of novel nanomedicines for cancer treatment”.  The authors have summarized the different aspects of polymers NPs, such as type of NPs, design, mechanism of action, methods of preparation. Further, they have also provided brief information about the clinically available polymeric NPs. Figure 1 is an excellent way of representation and timely mannered.

Response: We would like to thank the reviewer for his/her comments and revision of the manuscript.

Although the authors have shed a light on different aspects of polymeric NPs, still there a scope to increase the elegance of the manuscript.

Here are some of the suggestions.

I am wondering why the authors have used this abbreviation “therapeutic delivery systems (DDS)” instead of “drug delivery system (DDS)”. Either the authors should use TDS or DDS with respective terminology.

Response: we have changed drug delivery systems to therapeutic delivery systems throughout the manuscript in the revised manuscript.

Line 58-60: Not in a flow; please rephrase it.

Response: The paragraph has been rewritten.

Line 68: These two references would fit here;

https://doi.org/10.1080/09205063.2021.1909412

https://doi.org/10.1016/j.nantod.2020.101051

Response: We agree with the reviewer and both references have been added in the revised version.

Line 73: metabolites (not compounds)

Response: Corrected in the revised manuscript.

Line 98: production (not pro-duction)

Response: Corrected in the revised manuscript.

Figure 2:

In MOFs section- Expand “MOFs” in the legend; functionalization (not funcionalization)

Deleted excess “of” in MOFs and denrimers section

Replace “no” with “less” (eg., less biodegradable, less biocompatible, etc)

In metalic NPs section –  “thermal” (not termal)

Response: the Figure 2 in the revised manuscript has been corrected according to the typos identified. We really appreciate the revision.

Section mechanism of action of polymeric NPs - Please change word Phase with Stage. As in the next section the authors discussed the clinical trails readers may confuse with this terminology.

Response: we have performed that change.

Define the Fick’s law, Korsmeyer and peppas model, and Higuchi equation.

Response: the definition of Fick’s law, Korsmeyer and peppas model, and Higuchi equation has been explained in the revised manuscript. We thank the reviewer for this suggestion.

Line 223: if (not ff)

Response: the typo has been corrected in the revised manuscript.

Methods section - Although they are fundamental methods, they are the real working methods on the bench. Therefore, it would be interesting to illustrate each technique in a pictorial representation either from copyrighted or own designs. This would attract many budding researchers.

Response: We have designed a new figure (figure 7) of that illustrates the most significant techniques in polymeric formulation.

Section “polymeric NPs in clinical investigations” – Irrespective of the text, a table with the enlist of drug and polymer information would be attractive and more informative.

Response: We appreciate that comment, the table has been created.

Reviewer 5 Report

This review article provides interesting information and illustrations. However some minor revisions would improve its impact:  

Figures: A complete legend, which explains the figures and abbreviations, is missing and should be provided.

An additional table showing the advantages of nanoparticles compared to conventional chemotherapeutics is missing and should be provided.

Text: Polemics should be avoided, e.g. "People still die of cancer" implicates that nanoparticles may solve the cancer problem. That is unfortunately not the case.

The style of the review article should be revised by an experienced senior scientist and a native English speaker.

Author Response

This review article provides interesting information and illustrations. However some minor revisions would improve its impact:  

Response: We would like to thank the reviewer for the comments and suggestions.

Figures: A complete legend, which explains the figures and abbreviations, is missing and should be provided.

Response: We have added a more detailed description of each figure.

An additional table showing the advantages of nanoparticles compared to conventional chemotherapeutics is missing and should be provided.

Response: This suggestion has also been mentioned by other reviewers, and therefore we have added a table containing all this information.

Text: Polemics should be avoided, e.g. "People still die of cancer" implicates that nanoparticles may solve the cancer problem. That is unfortunately not the case.

Response: We fully agree with the reviewer and the sentence has been removed.

The style of the review article should be revised by an experienced senior scientist and a native English speaker

Response: We appreciate this comment. We have reviewed extensive the text in order to guarantee a high-quality stile for scientific English.   

Round 2

Reviewer 2 Report

The authors corrected the paper in an appropriate way. In my opinion, the manuscript is suitable for publication

Reviewer 4 Report

The authors have made substantial edits by considering the reviewer's comments. The revised manuscript is acceptable for publication.